# Seasonal Changes and the Interaction between the Horse Chestnut Leaf Miner *Cameraria ohridella* and Horse Chestnut Leaf Blotch Disease Caused by *Guignardia aesculi*

**Michal Kopačka** [1], **Gösta Nachman** [2] and **Rostislav Zemek** [1],*

[1]  Institute of Entomology, Biology Centre CAS, Branišovská 1160/31, 370 05 České Budějovice, Czech Republic; michalkopacka@seznam.cz

[2]  Department of Biology, Section of Ecology and Evolution, University of Copenhagen, 2100 Copenhagen, Denmark; gnachman@bio.ku.dk

*   Correspondence: rosta@entu.cas.cz

**Abstract:** The horse chestnut leaf miner *Cameraria ohridella* (Lepidoptera: Gracillariidae) is an invasive pest of horse chestnut *Aesculus hippocastanum* (Sapindales: Sapindaceae) and has spread through Europe since 1985. Horse chestnut leaf blotch is a fungal disease caused by *Guignardia aesculi* (Botryosphaeriales: Botryosphaeriaceae) that also seriously damages horse chestnut trees in Europe. The interaction between the leaf miner and the fungus has not yet been sufficiently described. Therefore, the aim of the present study was to assess leaf damage inflicted to horse chestnut by both *C. ohridella* and *G. aesculi* during the vegetation season and to model their interaction. The damage to leaf area was measured monthly from May to September 2013 in České Budějovice, the Czech Republic using digital image analysis of sampled leaves. A simple phenomenological model describing the expected dynamics of the two species was developed. The study revealed that the damage caused by both the pests and the fungus varied significantly among sampling sites within the city. The overall leaf damage exceeded 50% in no-raking sites in August. The mathematical model indicates that infestation by *C. ohridella* is more affected by *G. aesculi* than vice versa. *Guignardia aesculi* is thus the superior competitor of the two species. Our findings highlight the delicate interplay between insect pests and fungal pathogens and the spatiotemporal dynamics influencing them, calling for more research in this understudied area.

**Keywords:** *Aesculus hippocastanum*; insect pest; fungal disease; invasive species; leaf damage; model; competition; urban environment

## 1. Introduction

The horse chestnut tree *Aesculus hippocastanum* L. (Sapindales: Sapindaceae) is a deciduous tree widespread in European countries. This species originated in the Epirus region and the foothills of the Pindus Mountains in northwestern Greece and since the 17th century it has been planted in Europe as an ornamental tree [1]. Horse chestnut trees are frequently grown in city parks where they provide shade and reduce the urban heat island effect. *Aesculus hippocastanum* is also an important source of pollen and nectar for pollinating insects [2] and a habitat for many organisms including natural enemies, e.g., phytoseiid predatory mites [3–5].

The tree is attacked by an invasive pest, the horse chestnut leaf miner *Cameraria ohridella* Deschka and Dimic (Lepidoptera: Gracillariidae) which was described for the first time in 1985 near Lake Ohrid where it occurred in the area between Macedonia and Albania [6] and which probably represents the origin of this species [7]. The moth spread to all countries of Europe in less than three decades. In the Czech Republic, it was first recorded in 1993 and in the course of five years, it spread across the whole country [8,9]. Since 2004, horse chestnut leaf miner has also spread in Asia Minor [8,10,11]. Though

the main host tree is horse chestnut, the leaf miner occasionally attacks sycamore maple (*Acer pseudoplatanus* L.) [12]. *Cameraria ohridella* usually has three to four overlapping generations during the whole vegetation period. The number of generations per year depends on the temperature conditions [13,14]. The feeding activity of *C. ohridella* larvae causes the accumulation of phenolic compounds in tissues located at the border of mines, the desiccation of the adjacent leaf parts and the death of the epidermis on both sides of the leaf [15,16]. Heavily infested trees suffer from premature defoliation, which in turn reduces photosynthesis and, as consequence, negatively affects seed weight [17–19].

Horse chestnut leaf blotch is a fungal disease caused by *Guignardia aesculi* (Peck) V.B. Stewart (Botryosphaeriales: Botryosphaeriaceae) that is widely spread in North America, Asia, and Europe. *Guignardia aesculi* dispersed in Europe and probably arrived in the Czech Republic in the 1950s [20–22]. The fungus is in an asexual stage during the vegetation season when it parasitizes horse chestnut leaves, thereby reducing their assimilation efficiency, while it overwinters in its sexual stage as a saprophyte on fallen leaves [23,24]. *Guignardia aesculi* infects many species of the genus *Aesculus* [20,25,26]. The first stage of *G. aesculi* infection is water-soaked irregular areas on the epidermis layer. This symptom can be observed in Central Europe at the beginning of April, whereupon it spreads rapidly [27]. High air humidity promotes leaf infection by *G. aesculi* [28]. Damage to the leaf area during the vegetation season is obvious but it does not seem to significantly impair the overall health of horse chestnut trees [20,23,24].

Various methods to control the horse chestnut leaf miner and the leaf blotch have been investigated. Besides the application of insecticides by spray or trunk injection [29–32], several studies on biological control of *C. ohridella* using birds [33], parasitoids [34,35], entomopathogenic fungi [36–43] or entomopathogenic nematodes [44] have been reported. The mycoparasitic fungus *Trichoderma harzianum* Rifai has recently been found to be a promising biocontrol agent against the leaf blotch pathogen [45]. Nevertheless, the removal of fallen leaves of *A. hippocastanum* in which both the diapausing pupae of *C. ohridella* and the spermogonia stage of *G. aesculi* overwinter is considered to be an effective, economically and ecologically acceptable method of reducing both organisms [20,24,46,47].

The spread of horse chestnut leaf miner and its population dynamics, as well as the role of natural enemies, are relatively well described. The dynamics of damage inflicted by *G. aesculi* in urban environments has, however, not yet been sufficiently explained, nor has its interaction with *C. ohridella* [48]. Studies dealing with this type of interaction do not explicitly incorporate time into the statistical analyses of factors affecting levels of leaf damage [49,50]. A few studies have focused on the role played by chemical substances in the interactions between *C. ohridella* and *G. aesculi* [51–53]. Thus, Johne et al. [53] carried out laboratory experiments to investigate whether synthetic volatile organic compounds (VOCs), found to be emitted by *G. aesculi*-infected horse chestnut leaves, has an effect on the ovipositing behaviour of the leaf miner. Jagiełło et al. [51] used *A. hippocastanum* saplings to reveal the role of horse chestnut secondary metabolites in the host-pathogen-pest interaction. Later, these authors studied the effect of *G. aesculi* on the preference of *C. ohridella* with respect to oviposition under laboratory and greenhouse conditions [52].

The aim of the present study was to investigate to what extent a simple phenomenological model can be used to describe the dynamics of damage inflicted simultaneously by *G. aesculi* and *C. ohridella* to leaves of fully grown horse chestnut trees in an urban area. The model is used to provide insight into how the two pest species affect leaf damage over the vegetation season as well as how *G. aesculi* and *C. ohridella* interact with each other. Furthermore, we consider the possibilities of using the model as a predictive tool for pest management.

## 2. Materials and Methods

### 2.1. Study Site

The survey was conducted in the town of České Budějovice, South Bohemia (48° 59′ N, 14° 29′ E, Czech Republic). The total cadastral area is 56 km$^2$, and the total green public

open space is 260 ha inside the town. České Budějovice is the largest city in South Bohemia, with almost 100,000 inhabitants. The town is surrounded by forests and agricultural fields. The total number of horse chestnut trees in the town was estimated as 534 (Kopačka 2011). For the purpose of the present study, the city area was divided into eight semi-isolated sites (Table 1). All parameters and the map of the study sites are the same as those in a previous study [54], with the exception that two additional study sites, "Stromovka Park" and "Třebotovice and Kaliště village" were included in the survey. The area of green public open space at both study sites was one ha. The range of tree age was 23–37 years in Třebotovice and Kaliště village and 31–56 years in Stromovka Park. Only the Stromovka Park site is situated near a wetland, providing suitable conditions for the development of *G. aesculi.* All horse chestnut trees grow within 12–29 m of the wetland borders. Two rivers and some streams flow through České Budějovice, as seen from the map. Previous studies [28,54] have demonstrated higher damage to leaves attacked by *G. aesculi* in the city center than in other areas. At the Nádražní Street site, leaf litter is not removed; therefore, *A. hippocastanum* leaves are severely infested with *C. ohridella* every year [54]. The average cadastral area per site was 107.25 ha (SEM = 35.44, *n* = 8), and the average of green public open space was 21.55 ha (SEM = 7.43, *n* = 8). The average number of horse chestnut trees was 50.13 (SEM = 20.58, *n* = 8), and the average age of the studied trees was 52.13 years (SEM = 7.60, *n* = 8) at the study sites.

**Table 1.** Characteristics of the studied sites.

| Study Site | | Cadastral Area | Green Public Open Space | Horse Chestnut Trees | Avg. Age $\pm$ SEM [&] of Horse Chestnut Tree |
|---|---|---|---|---|---|
| Label | Name | (Hectares) | (Hectares) | (Number) | (Year) |
| A | City centre | 133 | 12 | 186 | 74.99 $\pm$ 1.86 |
| B | Šumava and Máj estate | 183 | 58 | 40 | 39.02 $\pm$ 3.19 |
| C | Vltava estate | 92 | 34 | 40 | 41.79 $\pm$ 3.49 |
| D | Třebotovice and Kaliště village | 30 | 2 | 5 | 17. 82 $\pm$ 6.86 |
| E | Rožnov estate | 294 | 29 | 57 | 70.49 $\pm$ 4.43 |
| F | Pražské předměstí estate | 124 | 37 | 49 | 49.74 $\pm$ 2.75 |
| G | Stromovka Park | 1 | 1 | 6 | 43.53 $\pm$ 4.13 |
| H | Nádražní Street | 1 | 1 | 18 | 77.74 $\pm$ 3.02 |

[&] SEM = standard error of the mean.

### 2.2. Sampling of Horse Chestnut Leaves

Leaf samples were collected at the eight sites five times during the whole vegetation season. The leaves from a given site were collected within the same day. Sampling did not take place within 48 hours after a rain event. The sampling dates were from 16 to 24 May, from 13 to 21 June, from 11 to 19 July, from 11 to 22 August and from 9 to 19 September. The standard sample of leaves from a site contained thirty randomly selected compound leaves from individual tree branches up to 2.5 m aboveground. The leaf stalk was only used to handle the leaf. After removal, individual leaves were put into plastic bags and immediately placed in a portable cool box to be transported to the laboratory, where the leaves were stored in a refrigerator at 4 °C for a maximum of 24 h.

### 2.3. Measurement of Proportion of Leaf Area Damaged by Leaf Miner and Fungus

The total leaf area and damaged area of the sampled leaves were measured by image analysis of digital photographs. First, the leaves were photographed in a temporary photo studio. Each compound leaf was laid on a white semimatte board (100 cm × 100 cm) with a measurement scale and grayscale calibration chart to be photographed by a Nikon D 5100 equipped with a lens Nikon AF-S NIKKOR 18–55 mm, 1: 3.5–5.6 G (Nikon Corporation, Tokyo, Japan). The camera was fastened vertically to a tripod placed 75 cm above the leaf. The leaf was lit by three halogen lamps (400 W each). One lamp was placed one meter behind the board to illuminate the backside of the leaf; two lamps were placed at

an angle of approximately 45 degrees above the semimatte board at a distance of 1.5 m and illuminated from the side of the leaf. It was necessary to limit other light sources to a minimum, and care was taken to fasten the camera in a stable position and maintain the same setting while taking photos of the whole series. Leaflets of a compound leaf were fixed so that they did not overlap. This enabled us to analyze images semiautomatically, as described below.

Pictures were taken at a high resolution of 3264×4928 pixels and saved in JPEG format (the average file size was approximately 2 MB). The images were digitally cleaned using Adobe Photoshop CS 3 (Adobe Inc., San Jose, California, USA). The main principle of the analysis was based on thresholds [55,56]; therefore, the contrast between the green leaf area and white background was first increased. Then, with the purpose of measuring the total leaf area, the leaf stalk and all possible impurities on the white background were painted with a white color. In order to measure leaf area damaged by *C. ohridella*, the area of leaf mines was repainted with a white color. This picture was saved before using it to assess the leaf area damaged by *G. aesculi*. The area destroyed by this fungus was repainted, and the resulting image was saved separately. Marking the area and repainting was performed manually by means of 3D SpaceNavigator™ model 3DX-700028 (3Dconnexion, Boston, Massachusetts, USA) and standard tools in Adobe Photoshop. Each sampled leaf thus resulted in three images amounting to 1200 threshold images during the entire study. These images were subsequently processed by our custom-made software written in Java™ programming language [57], which saved the values of the areas (number of black pixels in an image) in CSV files (comma-separated values). The damages attributed to *C. ohridella* and *G. aesculi* were finally calculated as percentages of the total leaf area.

### 2.4. Data Presentation and Statistical Analysis

All measured data stored in CSV files were imported into a Microsoft Access 2010 database. As the dependent variable (percentage area destroyed) was not normally distributed, the arcsine square-root transformation [58,59] was used to normalize data before the statistical analysis. Two-way multivariate analysis of variance (MANOVA) with the sampling site and period of sampling as the main factors was used to analyze the data. The analysis was performed using STATISTICA software (Version 13.2, StatSoft Inc., Tulsa, OK, USA).

### 2.5. Modeling Dynamics of Leaf Damage Caused by Cameraria ohridella and Guignardia aesculi

Since the interactions between *C. ohridella* and *G. aesculi* during a vegetation season may change as the proportions of damaged leaf area attributed to each species increase, we developed a simple phenomenological model describing the expected dynamics of the two species. As new leaves were sampled at each sampling occasion, we did not have sufficient information to validate the model against data obtained from individual leaves. It was therefore assumed that the processes taking place at individual leaves are reflected by the data obtained as average proportions of damage per sampling site assessed during the five sampling periods.

#### 2.5.1. The Model

The proportions of leaf area damaged by *C. ohridella* and *G. aesculi* at time *t* are denoted $p_C(t)$ and $p_G(t)$, respectively, where $0 \leq p_C(t) \leq 1$ and $0 \leq p_G(t) \leq 1$. Furthermore, as the total damaged leaf area at time *t*, found as $p_{total}(t) = p_C(t) + p_G(t)$, cannot exceed 1, we also have the constraint that $0 \leq p_{total}(t) \leq 1$.

If only a single species is present, the increase in the proportion of leaf area damaged by the species (*dp/dt*) is assumed to increase in the beginning of the season (when *p* is still small) and then to level off as *p* approaches 1. Such dynamics can be described mathematically by the well-known logistic growth model as:

$$\frac{dp}{dt} = \alpha p(1 - p) \tag{1}$$

where $\alpha$ is a parameter expressing the speed at which the species inflicts damage. Equation (1) can be generalized by introducing two parameters (called $\beta$ and $\gamma$), yielding

$$\frac{dp}{dt} = \alpha p^{\beta}(1-p)^{\gamma} \qquad (2)$$

where $\beta = \gamma = 1$ corresponds to the standard logistic equation (Equation (1)). As long as $p$ is close to 0, the initial growth rate is seen to be $\frac{dp}{dt} \approx \alpha p^{\beta}$, implying that damage will grow linearly with time for $\beta = 0$ (yielding $p(t) = p(0) + \alpha t$) and exponentially for $\beta = 1$ (yielding $p(t) = p(0)e^{\alpha t}$). For a given value of $p$, the growth rate is seen to decline with increasing values of $\beta$ (because $p < 1$). This will still lead to exponential growth but will prolong the lag phase before $p$ reaches a level at which growth starts to be inhibited by damage. This happens when the term $(1-p)^{\gamma}$ gradually declines from being close to 1 to approaching 0 when $p$ approaches 1. How fast this occurs depends on the parameter $\gamma$. Thus, if $\gamma = 0$, the species will grow exponentially until $p$ becomes 1, whereas growth will level off at lower values of $p$, the higher the value of $\gamma$. In other words, $\gamma$ expresses how sensitive the growth of a species is to damage.

We assume that the growth rate of a species is not only inhibited by its own damage, but also by the damage caused by other species. It means that the growth rate of the $i$th species can be written as:

$$\frac{dp_i}{dt} = \alpha_i p_i^{\beta_i}(1 - p_{total})^{\gamma_i} \qquad (3)$$

where $p_i$ is the proportion of damage caused by species $i$, while $\alpha_i$, $\beta_i$ and $\gamma_i$ are species-specific values of $\alpha$, $\beta$ and $\gamma$.

When the parameters of Equation (3) have been estimated (see below), we can use the model to estimate the mutual effect of the two species on each other by comparing the damage each species inflicts when it is either alone or together with the other species. The relative impact imposed by the competitor species on species $i$ at time $t$ is found as:

$$RI_i(t) = \frac{p_i'(t) - p_i(t)}{p_i'(t)} \qquad (4)$$

where $p_i(t)$ is the damage caused by species $i$ when it coexists with its competitor, while $p_i'(t)$ is the predicted damage when it occurs alone, i.e., by setting $p_{total} = p_i$ in Equation (3).

### 2.5.2. Estimation of Model Parameter Values

The predicted damages caused by the two species were obtained from Equation (3) by replacing $dp_i/dt$ with $\Delta p_i/\Delta t$, using a time step ($\Delta t$) of a half day starting from time $t = 0$. Thus, the predicted damage attributed to species $i$ at time $t + \Delta t$ was calculated as $\hat{p}_i(t + \Delta t) = \hat{p}_i(t) + \Delta \hat{p}_i$. $t$ was calculated as the number of days since the first sampling. As sampling was conducted over several days, we used the midpoints of each sampling period to calculate $t$, implying that $t = 0$ corresponds to May 20. Thus, for the following four sampling periods, $t$ was set to 28, 56, 85.5, and 117 days, respectively.

We assumed that the underlying biological processes taking place at the eight sampling sites were the same, implying that the values of the model's six parameters were independent of the site, in contrast to the 16 values of $p_i(0)$, which were estimated for each site and species. In total, this yielded 22 unknown constants that had to be estimated from 80 data points (two assessments of damage obtained at eight sampling sites on five sampling occasions). For this purpose, we used the Solver tool in Excel®. The method is based on iterating the model's constants until the sum of squared deviations (SSD) between the observed and estimated values of $p_i(t)$ is minimized. Ideally, the values of the constants should gradually converge from their initial values (chosen by the researcher) towards a final set of values, which represents the best estimates of the model's constants (i.e., the values that minimize SSD). However, in order to reduce the risk of ending up in a local minimum, it is recommended to start the iteration procedure with different combinations

of initial values and then check whether the procedure converges to the same set of values, irrespective of their initial values.

The quality of a fit was expressed as the proportion of total variation in experimental data that could be explained by the model ($R^2$), while the significance of the $R^2$ was assessed by an *F*-test as the ratio between the mean square (MS) of the model divided by the mean square error (MSE) with $p - 1$ degrees of freedom (df) in the numerator and $n - p$ df in the denominator. $p$ is the number of estimated values (22) and $n$ the number of data points (80).

The six parameter values estimated above were used to model the progress in overall damage calculated as the average of the eight sampling sites. This reduced the number of constants to two (i.e., one estimate of $p(0)$ for each species), which were estimated by fitting the model to 10 data points. The resulting model was used to evaluate the relative impact the two pest organisms have on each other.

### 2.5.3. Evaluating the Model as a Predictive Tool

We also examined whether the model could serve as a tool for early warnings based on data sampled in the beginning of the season. Since the mathematical model is deterministic and sensitive to the initial state of the system, two conditions have to be satisfied in order to use the model for tactical purposes: (i) the sample-based estimates of $p_i(0)$ (denoted $\overline{p}_i$) should be close to the a posteriori values (denoted $\hat{p}_i$), obtained retrospectively by fitting the model to data spanning over the entire season; and (ii) the biological system should basically be predictable with relatively little noise obscuring the underlying dynamics described by the model.

The first condition was tested by means of linear regression, where values of $\overline{p}_i$ were regressed against the corresponding values of $\hat{p}_i$, yielding eight data points per species. The condition was met if the line showed a significant positive correlation with a slope close to 1 and an intercept close to 0 so that $\overline{p}_i = \hat{p}_i$. The second condition was tested by calculating the correlation coefficients between the damage assessed at sampling $k$ ($k$ = 1,2,3,4) and the final damage assessed at the fifth sampling, using pairwise data from the eight sampling sites and for each species separately. The condition was met if the correlation coefficient ($r$) between damages assessed at two sampling occasions was significantly greater than 0.

## 3. Results

### 3.1. Damage to A. hippocastanum Leaves during the Vegetation Period

Changes in the average percentage of damage and the range of damaged leaf area inflicted by *C. ohridella* and *G. aesculi* across all sites during the vegetation season indicated a cumulative pattern (Table 2). The average damaged leaf area varied highly among the study sites (Figure 1, data points). The overall average percentage of leaf area destroyed by *C. ohridella* and *G. aesculi* was 3.06% (SEM = 0.23, *n* = 1200) and 2.75% (SEM = 0.21, *n* = 1200), respectively. Of the sites, Nádražní Street had the highest leaf damage inflicted by *C. ohridella*, reaching 78.79%. The highest leaf damage inflicted by fungal pathogens was recorded at the Stromovka Park site and reached 78.86%. Both values were recorded in September.

**Table 2.** The average percentage of horse chestnut leaf area damaged by *Cameraria ohridella* and *Guignardia aesculi* during vegetation season. Pooled data from all sampling sites across city (average ± SEM, *n* = 240).

| Sampling Date | Leaf Area Damaged by *Cameraria ohridella* | | Leaf Area Damaged by *Guinardia aesculi* | |
|---|---|---|---|---|
| | Avg. ± SEM | Range | Avg. ± SEM | Range |
| from 16 to 24 May | 0.018 ± 0.002 | 0.000–0.329 | 0.031 ± 0.005 | 0.000–0.591 |
| from 13 to 21 June | 0.214 ± 0.117 | 0.000–28.071 | 0.448 ± 0.126 | 0.000–27.612 |
| from 11 to 19 July | 2.594 ± 0.402 | 0.003–37.582 | 0.924 ± 0.095 | 0.000–14.882 |
| from 11 to 22 August | 5.058 ± 0.756 | 0.000–78.790 | 2.726 ± 0.255 | 0.022–22.636 |
| from 9 to 19 September | 7.416 ± 0.666 | 0.006–60.686 | 9.612 ± 0.891 | 0.000–78.864 |

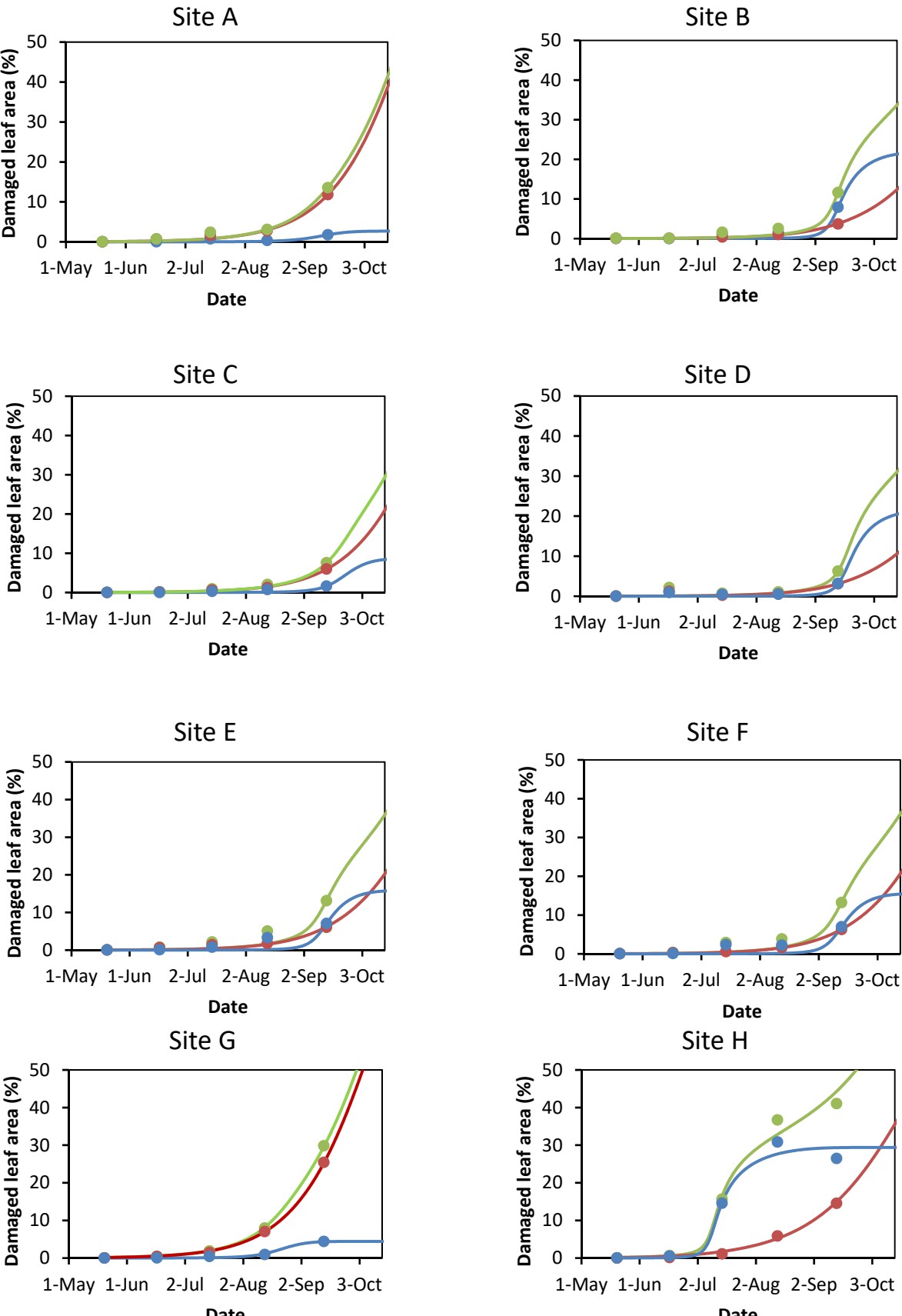

**Figure 1.** The observed (points) and model predicted (lines) damages during one season at the eight sampling sites (Table 1). First sampling took place around May 20. Each point represents the average damage of 30 leaves attributed to *C. ohridella* (**blue**) and *G. aesculi* (**red**) or in combination (**green**).

The MANOVA revealed a highly significant effect of both the site (Wilk's $\lambda$ = 0.428, $F_{14,2318}$ = 87.532, $p < 0.0001$) and sampling period (Wilk's $\lambda$ = 0.199, $F_{8,2318}$ = 359.882, $p < 0.0001$) on leaf damage. The interaction between the site and period was also highly significant (Wilk's $\lambda$ = 0.629, $F_{56,2318}$ = 10.813, $p < 0.0001$).

Consecutive ANOVAs confirmed a highly significant effect of both the site and sampling date on damage caused by *C. ohridella* ($F_{7,1160}$ = 156.853, $p < 0.0001$ and $F_{4,1160}$ = 556.997, $p < 0.0001$, respectively). The interaction between these two factors was also highly significant ($F_{28,1160}$=13.604, $p < 0.0001$). A highly significant effect of both site and sampling date was also confirmed for *G. aesculi* damage ($F_{7,1160}$ = 33.618, $p < 0.0001$ and $F_{4,1160}$ = 466.230, $p < 0.0001$, respectively). The interaction between these two factors was also highly significant ($F_{28,1160}$ = 8.082, $p < 0.0001$).

### 3.2. Modeling Leaf Damage Caused by Cameraria ohridella and Guignardia aesculi

Figure 1 shows that the model fitted the data very well at all sampling sites ($R^2$ = 0.972; $F_{21,58}$ = 94.4; $p < 0.0001$). The estimated parameter values and the initial damages due to *C. ohridella* and *G. aesculi* are shown in Table 3.

**Table 3.** Estimated parameter values and site-specific initial damages (proportions). $\alpha$ has dimension $d^{-1}$, while $\beta$ and $\gamma$ are dimensionless.

|  |  | Species | |
| --- | --- | --- | --- |
|  |  | *C. ohridella* | *G. aesculi* |
| Parameters | $\alpha$ | 3.0066 | 0.0478 |
|  | $\beta$ | 1.5023 | 1.0096 |
|  | $\gamma$ | 14.611 | 0.6732 |
| Initial damage ($p(0)$) | Site A | $5.42 \times 10^{-5}$ | $6.54 \times 10^{-4}$ |
|  | Site B | $4.35 \times 10^{-5}$ | $2.11 \times 10^{-4}$ |
|  | Site C | $4.30 \times 10^{-5}$ | $3.25 \times 10^{-4}$ |
|  | Site D | $3.96 \times 10^{-5}$ | $1.73 \times 10^{-4}$ |
|  | Site E | $4.82 \times 10^{-5}$ | $3.44 \times 10^{-4}$ |
|  | Site F | $4.84 \times 10^{-5}$ | $3.51 \times 10^{-4}$ |
|  | Site G | $8.96 \times 10^{-5}$ | $1.538 \times 10^{-3}$ |
|  | Site H | $2.29 \times 10^{-4}$ | $1.453 \times 10^{-3}$ |

Figure 2 shows the fit of the model to the overall development in damage averaged over the eight sites. The parameter values are the same as in Table 1, so only the initial proportions of damage were estimated as $5.489 \cdot 10^{-5}$ (0.0055%) and $5.278 \cdot 10^{-4}$ (0.0528%) for *C. ohridella* and *G. aesculi*, respectively. The fit was highly significant ($R^2$ = 0.983; $F_{1,8}$ = 469.7; $p < 0.0001$).

We may use the parameter values given in Table 3 to estimate the growth rates in the beginning of the season (i.e., when $p \approx 0$). This yields $\frac{dp}{dt} = \alpha p^{\beta} = 3.0066 \times p^{1.5034} d^{-1}$ for *C. ohridella* and $0.0478 \times p^{1.0096} d^{-1}$ for *G. aesculi*. Thus, for a given value of $p$, the growth rate of *C. ohridella* will exceed that of *G. aesculi*, provided $p > 0.00023$.

On the other hand, the high value of $\gamma$ indicates that *C. ohridella* is more inhibited by damage inflicted by both itself and its competitor than *G. aesculi* is. Thus, the model predicts that damage due to *C. ohridella*, even in the absence of *G. aesculi*, will level off and not exceed 50%. In contrast, *G. aesculi* seems rather unaffected by the presence of *C. ohridella* and may cause close to 100% damage if the season is long enough. As seen from Figure 3, the relative impact of *G. aesculi* on *C. ohridella* reaches more than 70% while that of *C. ohridella* on *G. aesculi* is only about 10%.

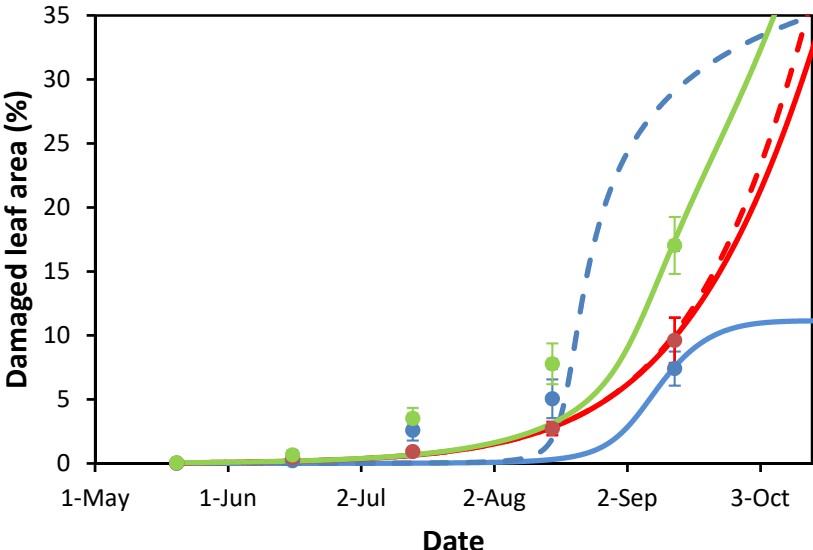

**Figure 2.** Damage caused by *C. ohridella* (**blue**) and *G. aesculi* (**red**) or combined (**green**) averaged over the eight sampling sites so that each dot represents the average (±2SEM) of 240 leaves. Full lines show the damage when both species are present, while the dashed lines show the expected damage of a species in absence of the other. The lines were calculated by means of Equation (3) using the values of $\alpha$, $\beta$ and $\gamma$ given in Table 3, and with $p(0)$ equal to $5.489 \times 10^{-5}$ for *C. ohridella* and $5.278 \cdot 10^{-4}$ for *G. aesculi*.

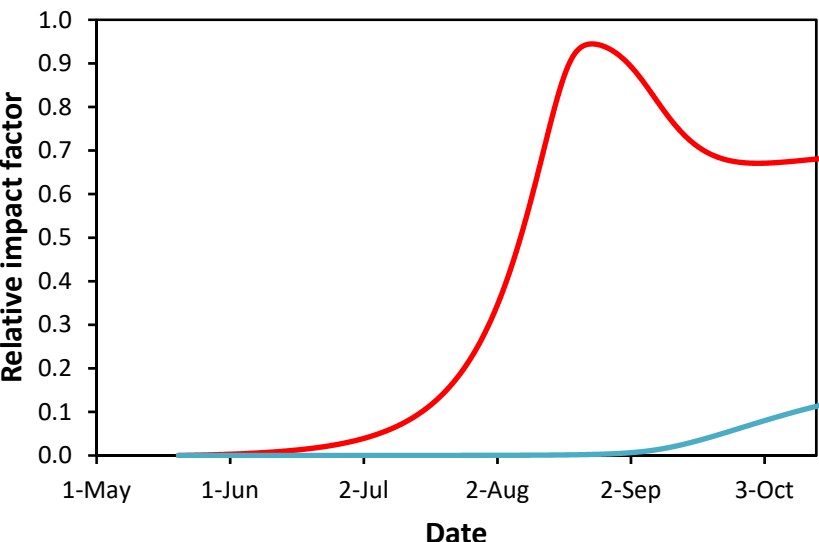

**Figure 3.** Relative impact of the two species on each other based on Figure 2. Blue line: impact of *C. ohridella on G. aesculi*. Red line: impact of *G. aesculi on C. ohridella*.

### 3.3. The Model as a Predictive Tool

Though the model in retrospect provided a good fit to seasonal data, its capacity as a tactical tool for long-term forecasts of leaf damage seems limited by the fact that there was no agreement between values of $\hat{p}_i$ and $\overline{p}_i$ (*C. ohridella*: $\overline{p}_i = 0.1749\hat{p}_i + 0.0163$; $r = 0.2668$; $F_{1,6} = 0.460$; $p = 0.523$; *G. aesculi*: $\overline{p}_i = -0.0896\hat{p}_i + 0.0363$; $r = -0.3487$; $F_{1,6} = 0.830$; $p = 0.3973$). In other words, if the estimates of $p_i$ obtained by leaf sampling in mid-May had been used to initialize the model, the predicted dynamics would differ significantly from what was actually observed.

This finding is confirmed by the fact that damages assessed from the first sampling (mid-May) were poorly correlated with the final damages assessed in mid-September

(Table 4). The same applies to data collected in mid-June, whereas data collected in mid-July and mid-August were more strongly correlated with the final damages.

**Table 4.** Correlations between damages caused by *C. ohridella* and *G. aesculi* assessed at two different sampling occasions.

| Periods | *C. ohridella* | | | *G. aesculi* | | |
|---|---|---|---|---|---|---|
| | *r* | $F_{1,6}$ | *p* | *r* | $F_{1,6}$ | *p* |
| Period 1 vs. Period 5 | 0.3975 | 1.126 | 0.3295 | −0.1823 | 0.206 | 0.6657 |
| Period 2 vs. Period 5 | −0.2950 | 0.572 | 0.4781 | −0.1429 | 0.125 | 0.7358 |
| Period 3 vs. Period 5 | 0.9698 | 95.02 | <0.0001 | 0.6848 | 5.299 | 0.0609 |
| Period 4 vs. Period 5 | 0.9715 | 100.7 | <0.0001 | 0.9555 | 62.96 | 0.0002 |

## 4. Discussion

### 4.1. Damage to the Leaf Area of A. hippocastanum during the Vegetation Period

Regarding *C. ohridella*, very little damage was found in the first sampling period in May, while one month later, the amount of damage was ten times higher. This finding is consistent with the results of a previous study reporting that the first generation of horse chestnut leaf miners started to hatch at the end of April [54]. The diapausing pupas of *C. ohridella* overwinter on fallen leaves. Therefore, the highest population density of the horse chestnut leaf miner is expected at sites where no raking takes place [46,60]. This was the case for the Nádražní Street site, where damage reached the highest level compared to the other sites in mid-July, and this trend persisted for the remainder of the vegetation season. The study by Salleo et al. [19] revealed that serious damage to the leaf area causes premature tree defoliation. The combined damage to the leaves exceeded 50% at the no-raking sites in August. The last sampling in September revealed less damage, which might be because the most damaged leaves had already fallen off.

The first stage of *G. aesculi*, characterized by water-soaked irregular areas on the epidermis layer of compound leaves, had been recorded in Slovakia from the beginning of April, followed by a rapid spread of this symptom [27]. The occurrence of the first symptoms in April was also indicated by our study because the necrotic leaf area of horse chestnut was observed in mid-May. The spermogonia stage of *G. aesculi* overwinters on fallen leaves of horse chestnut; therefore, the Nádražní Street site should have been the most infected with *G. aesculi* [20,24]. However, the increased incidence of leaves infected by *G. aesculi* at this site in spring was not obvious. Our previous study [28] indicated that high air humidity promotes infection by *G. aesculi*. This finding was also confirmed in the present study. In August, the greatest leaf area damaged by *G. aesculi* was recorded at three sites: the city center, where many trees grow in parks along two rivers; the Nádražní Street site that was not mowed, which might have contributed to an increase in air humidity; and the Stromovka Park site, where the trees grow near wetlands. In September, Stromovka Park began to have the greatest damaged leaf area due to *G. aesculi,* and this trend remained until the end of the vegetation period. The average damaged leaf area caused by *G. aesculi* was found to be 3.9% in Rimavská Sobota Park in Slovakia in September [27]. In comparison, the average leaf area damaged by *G. aesculi* found in the present study was 9.6%.

### 4.2. Interaction between C. ohridella and G. aesculi on the Leaves of Horse Chestnut Trees

According to Hatcher [61], the relationship between herbivorous insects and fungal diseases is often competitive, as they share the same resource. Surveys conducted in Bern in September by Gilbert et al. [49] reported that the occurrence of horse chestnut leaf blotch estimated on a score between 0 (not present) and 4 (very high) was inversely related to infestation by the horse chestnut leaf miner. The mechanisms of interaction between these two species have not yet been fully elucidated.

The model presented in this study aimed at explaining the underlying dynamics of two pest organisms (*C. ohridella* and *G. aesculi*) simultaneously attacking leaves of horse

chestnut trees. The model has only two state variables, representing the current proportions of leaf area damaged by each species, and three parameters per species (denoted $\alpha$, $\beta$ and $\gamma$). $\alpha$ represents the relative damage rate (increase in proportional damage per day), $\beta$ expresses whether the damage rate depends on how much area the species has already destroyed, and $\gamma$ expresses how fast the damage rate declines as the total damage progresses. As seen from Table 3, $\alpha$ for *C. ohridella* was estimated to be more than 60 times higher than for *G. aesculi*, indicating that the former species has the potential to become a serious pest. On the other hand, $\gamma$ for *G. aesculi* was found to be much smaller than that of *C. ohridella*, indicating that *G. aesculi* can achieve higher levels of damage than *C. ohridella*. Both species have values of $\beta$ values close to unity, which indicates that the damage rate is accelerated the more damage a species has already inflicted. This seems reasonable for *G. aesculi* since the production of spores is likely to increase proportionally with the infected area. For *C. ohridella*, the explanation for the positive $\beta$ is likely to be a combination of recruitment of young larvae, an increasing number of mines due to overlapping generations [13,14], and a positive feedback between the body size and the feeding rate of individual larvae. As the larvae grow, their larger size will speed up their total damage rate, even if the number of larvae per leaf declines due to pupation and mortality. When the larvae reach their final body size, they stop feeding and pupate. Studies have shown that the proportion of pupae entering diapause is gradually increasing during high-summer [14,62], which may explain why the increase in observed damage ceases from around mid-August. In contrast, *G. aesculi* continues to destroy the leaf area throughout the entire vegetation season (Figures 1 and 2).

*Cameraria ohridella* is expected to reach higher levels of damage in the absence of *G. aesculi* compared to when the fungus is present (Figure 2), whereas the effect of *C. ohridella* on *G. aesculi* is rather small (Figure 3). This result contrasts with the findings of a previous study [51] where the authors measured the response of horse chestnut saplings to separated and simultaneous colonization by *C. ohridella* and *G. aesculi*. The authors found that leaf damage increased faster when only the fungal pathogen attacked the plants than when they were infected by both pests. The authors concluded that simultaneous infestation by fungal and insect agents made the conditions unfavorable for the former species. The strong influence of *G. aesculi* on *C. ohridella* found in the present paper is attributed to the circumstance that feeding of *C. ohridella* larvae is hampered by a lack of suitable food, which may simultaneously decrease the per-capita feeding rate and increase the larval mortality. In contrast, *G. aesculi* is only limited by the amount of undamaged leaf area. However, resource competition may not be the only explanation for why *C. ohridella* is more sensitive to the presence of *G. aesculi* than vice versa. Thus, it cannot be excluded that interference competition may also play a role, e.g., if the fungus produces substances that are toxic or repellent to *C. ohridella*. The content of phenolic compounds, secondary metabolites produced by plant during defense response to pest or pathogen attack, was found to be higher in *A. hippocastanum* saplings on which both insect and leaf blotch disease were co-existing compared to a single-species infestation [51]. Evidence of a repellent role of volatile organic compounds (VOCs), which were found to be emitted by horse chestnut leaves in response to fungal infections by *G. aesculi* and/or powdery mildew *Erysiphe flexuosa* (Peck) Braun and Takam, was given by Johne et al. [53]. In contrast to these results, experiments by Jagiełło et al. [52] did not confirm that females of *C. ohridella* deposit eggs more frequently on healthy leaflets than on infected ones. A recent study demonstrated that the phyllosphere of *A. hippocastanum* is inhabited by many generalist endophytes, epiphytes and saprotrophic fungi, but that occurrences of common phyllosphere fungi were unrelated to the degree of damage caused by *C. ohridella* [63]. A positive interaction was, on the other hand, described between horse chestnut leaf miner and bleeding canker of horse chestnut (*Pseudomonas syringae* pv. *aesculi*), which is a serious bacterial disease that is lethal to *A. hippocastanum,* unlike *C. ohridella* and *G. aesculi*. In this case, feeding by *C. ohridella* larvae may cause the suppression of two defensive enzymes within wood tissue, thus facilitating bacterial infection [64,65]. We think that data based on repeated

observations of the same leaves throughout an entire season could help with deciding to what extent chemical defense mechanisms are involved in the competitive interactions, for instance, by studying whether *C. ohridella* establishes more readily on leaves free of *G. aesculi* than on leaves already infested by leaf blotch.

We investigated to what extent the strategic model presented in this paper can also be used as a predictive tool to forecast whether a stand of chestnut trees is at risk of suffering severe damage later in the season. Having a reliable forecast system would assist in deciding when and where treatment with pesticides will be needed. However, a crucial factor for making accurate predictions is that the current state of the system can be determined with little uncertainty and then used as an input in a model that predicts the system's state later in the season. Sampling error in itself may not be a serious problem as it can be solved by increasing the number of leaves sampled. A more serious problem is that the inherent predictability of the system seems to be low. Thus, we did not find evidence of a correlation between the damage assessed in mid-May and the damage assessed in mid-September, rendering long-term prognoses based on modeling unlikely. This is not surprising in view of the multitude of factors that may influence the relationship, but not incorporated in the model, such a temperature, precipitation, humidity, dispersal, natural enemies, etc. [66]. All these unknown factors add up as "noise", which is likely to override the underlying deterministic signal represented by the present model. To incorporate external factors in a predictive model would increase its complexity far beyond the one presented in this paper and may require a model that can cope with stochasticity in order to generate predictions in terms of probabilities rather than expectancies. On the other hand, since the level of damage assessed in mid-July was found to be a good predictor of final damage, decisions as to whether control measures should be implemented or not could be taken in mid-July at the earliest.

## 5. Conclusions

Our findings demonstrate that a leaf miner (*C. ohridella*) and a fungal pathogen (*G. aesculi*) can inflict substantial leaf damage to urban *A. hippocastanum*, leading to early defoliation and reducing the esthetical value or even vigour of this important ornamental tree. Therefore, efforts to prevent the spread of leaf miners and leaf blotch in urban areas should be enhanced through scientifically based management systems. These two pests compete for the same resource (i.e., leaf area) and our model revealed that *G. aesculi* had a high impact on *C. ohridella*, whereas the impact of *C. ohridella* on *G. aesculi* was small, indicating that *G. aesculi* is the superior competitor of the two species. Though the model fit the data very well, its predictive power is too low to constitute a reliable management tool. Further research, e.g., using manipulative experiments, is therefore needed to elucidate plant-mediated interactions between *C. ohridella* and *G. aesculi* and factors that may influence the relationship, such as abiotic conditions, dispersal, natural enemies, etc., which should be incorporated in a predictive model.

**Author Contributions:** Conceptualization, R.Z.; methodology, R.Z., M.K. and G.N.; sampling, image analysis and data curation, M.K.; modelling and visualization, G.N.; writing—original draft preparation, M.K. and G.N.; writing—review and editing, M.K., G.N. and R.Z. All authors have read and agreed to the published version of the manuscript.

**Funding:** This research received no external funding.

**Data Availability Statement:** The data presented in this study are available on request from the corresponding author.

**Acknowledgments:** This research was conducted with the institutional support RVO: 60077344. The authors thank Katarína Pastirčáková and anonymous reviewers for their valuable comments that helped to improve this manuscript.

**Conflicts of Interest:** The authors declare no conflict of interest.

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
