# Peer review of "Seasonal Changes and the Interaction between the Horse Chestnut Leaf Miner Cameraria ohridella and Horse Chestnut Leaf Blotch Disease Caused by Guignardia aesculi"

_forests, doi:10.3390/f12070952_

Round 1

Reviewer 1 Report

1. General Information

The paper presented involves an approach to the interaction between the horse chest nut leaf miner, Cameraria ohridella, and the horse chestnut leaf blotch, Guignardia aesculin, by means as a mathematical model, developed to answer the referred interaction as well as  other similar  situations.

2. Section by Section

2.1. - Introduction

This section is comprehensible and has a lot of recent bibliography to consolidate the affirmations made.

2.2. - Material and Methods

Overall Material and Methods are quite clear and detailed. This section is written in a very comprehensible way. However, the calculation of parameters a, b and g are not easy to understand, neither is the way how the values presented for each parameters were obtained.

 2.3. - Results

 Results are well presented; graphic component is interesting and gives a lot of information easy to understand.

2.4. - Discussion

The discussion is well supported in bibliographic references and explains clearly the results observed.

 3. Suggestions

Despite the interest of the presented paper, I suggest:

 3.1. Line 61

 I suggest you to give an example of a biological control method to make the text more interesting.

 3.2. Line 180

 I strongly suggest to improve the description of a, b, and g parameters calculation in order to allow follow-up the experience done.

Author Response

We appreciate positive review and suggestion for improvement of our manuscript. 

  1. General Information

The paper presented involves an approach to the interaction between the horse chest nut leaf miner, Cameraria ohridella, and the horse chestnut leaf blotch, Guignardia aesculin, by means as a mathematical model, developed to answer the referred interaction as well as  other similar  situations.

  1. Section by Section

2.1. - Introduction

This section is comprehensible and has a lot of recent bibliography to consolidate the affirmations made.

2.2. - Material and Methods

Overall Material and Methods are quite clear and detailed. This section is written in a very comprehensible way. However, the calculation of parameters a, b and g are not easy to understand, neither is the way how the values presented for each parameters were obtained.

Response: New text along with new equations was added to material and methods section to explain in details how the parameters were calculated.

 2.3. - Results

 Results are well presented; graphic component is interesting and gives a lot of information easy to understand.

2.4. - Discussion

The discussion is well supported in bibliographic references and explains clearly the results observed.

  1. Suggestions

Despite the interest of the presented paper, I suggest:

 3.1. Line 61

 I suggest you to give an example of a biological control method to make the text more interesting.

Response: Paragraph in Introduction on control of horse chestnut pests was enhanced with emphasize on biocontrol agents and relevat references were added.

 3.2. Line 180

 I strongly suggest to improve the description of a, b, and g parameters calculation in order to allow follow-up the experience done.

Response: Text of sections 2.5.1 and 2.5.2 on the model and parameters estimation was substantially revised/enhanced.

Reviewer 2 Report

Overall, the study is well conceived and thoroughly conducted.

The authors have presented the introduction, results, discussion and conclusion eloquently.

Minor concerns like:

In the title the author has written “chestnut leaf blotch Guignardia aesculin”, in my opinion the disease name shouldn’t be immediately followed by the pathogen name. The authors can say something like “chestnut leaf blotch pathogen Guignardia aesculin” or chestnut leaf blotch caused by Guignardia aesculin”. Same thing needs to be done in lines 12 and 48.

Lines 51-53 : Please review the sentence, I think it’s grammatically incorrect.

Line 383: “fenolic” is not the correct word, it should be replaced by “phenolic” .

Author Response

We appreciate positive review and suggestions for improvement of our manuscript.

Overall, the study is well conceived and thoroughly conducted.

The authors have presented the introduction, results, discussion and conclusion eloquently.

Minor concerns like:

In the title the author has written “chestnut leaf blotch Guignardia aesculin”, in my opinion the disease name shouldn’t be immediately followed by the pathogen name. The authors can say something like “chestnut leaf blotch pathogen Guignardia aesculin” or chestnut leaf blotch caused by Guignardia aesculin”. Same thing needs to be done in lines 12 and 48.

Response: The text was corrected to: “leaf blotch caused by Guignardia aesculi”. In addition, proper author name was written at the first occurrence of this species name in Introduction.

Lines 51-53 : Please review the sentence, I think it’s grammatically incorrect.

Response: The sentence was rewritten to: “The fungus is in an asexual stage during the vegetation season when it parasitizes the assimilation area of horse chestnut trees, while it overwinters in its sexual stage as a saprophyte on fallen leaves”.

Line 383: “fenolic” is not the correct word, it should be replaced by “phenolic” .

Response: This typo was corrected.